# A Wearable Patch Sensor for Simultaneous Detection of Dopamine and Glucose in Sweat

Yue Sun †, Junjie Ma †, Yuwei Wang, Sen Qiao, Yihao Feng, Zhanhong Li * , Zifeng Wang, Yutong Han and Zhigang Zhu

School of Health Science and Engineering, University of Shanghai for Science and Technology, Shanghai 200093, China; suny1717220@163.com (Y.S.); j_j_ma8883@163.com (J.M.); wangyuwei010@163.com (Y.W.); qs20030328@163.com (S.Q.); f36494426@163.com (Y.F.); zfwang@usst.edu.cn (Z.W.); yutonghan@usst.edu.cn (Y.H.); zhigang_zhu259@163.com (Z.Z.)

\* Correspondence: zhli@usst.edu.cn

† These authors contributed equally to this work.

**Abstract:** Achieving quantification of biomarkers in body fluids is crucial to the indication of the state of a person's body and health. Wearable sensors could offer a convenient, fast and painless sensing strategy. In this work, we fabricated a wearable electrochemical patch sensor for simultaneous detection of dopamine and glucose in sweat. The sensor was printed on a flexible PDMS substrate with a simple screen-printed method. This prepared four-electrode sensor integrated two working electrodes for dopamine and glucose electrochemical sensing, one Ag/AgCl reference electrode and one carbon counter electrode, respectively. Cyclic voltammetry, differential pulse voltammetry and chronoamperometry were used for the evaluation of the wearable electrochemical patch sensor. It exhibits good sensitivity, wide linear range, low limit of detection, good anti-interference and reproducibility toward dopamine and glucose sensing in PBS and sweat.

**Keywords:** dopamine; glucose; wearable; sensor; electrochemistry

## 1. Introduction

The concentration of biomarkers (glucose, dopamine, uric acid, $Na^+$, $H^+$, etc.) in body fluids is a key indicator of the state of a person's body and health. E.g., the concentration of glucose in blood is a gold standard for glucose metabolism disorder evaluation. Chronic high blood glucose level is usually associated with diabetes, yet low blood glucose level is associated with hypoglycemia [1]. Dopamine is an important neurotransmitter. Several nervous system diseases, such as Parkinson's disease, dementia with Lewy bodies, attention-deficit/hyperactivity disorder, and schizophrenia, are associated with dysfunctions of dopamine [2].

Traditional biomarker detection methods, including gas chromatography, high-performance liquid chromatography, enzyme-linked immunosorbent assay, and so on, are time consuming, costly, requiring complex equipment and professional operators. Electrochemical sensing method offers a compact, low-cost, convenient and fast strategy for biomarker detection. Plenty of efforts, including those of our research group, have been devoted to developing various electrochemical biomarker sensors [3–6]. In recent decades, the field of research and applications of wearable electrochemical sensing devices in health and environment monitoring has been boosting [7–9]. Biomarkers in sweat exhibit some degree of correlation with components in blood [10,11], and there is a lot of excellent works trying to detect biomarkers electrochemically in sweat as an alternative to blood testing [12–15], such as the wearable electrochemical devices for glucose [16,17], uric acid [18,19], lactate [20–22], dopamine [23,24], cortisol [25] and electrolyte [26,27] sensing. Dual-function or multi-function electrochemical sensing devices can provide richer sweat biomarker

information, and many works have achieved good results [22,28–30]. Overall, many efforts have been made in developing diverse wearable electrochemical sensing devices for biomarkers in sweat, and they have great application potential in realizing convenient biomarker detection and reducing the psychological burden of the subjects who need to pierce the skin for blood collection. Thus, a sensor capable of simultaneous electrochemical detection of dopamine and glucose can be a promising tool for the health management and diagnosis of these two-compound-related diseases. However, to the best of our knowledge, there is still no report on a wearable electrochemical sensor for the simultaneous detection of dopamine and glucose in sweat.

In this work, we fabricated a wearable patch sensor for simultaneous electrochemical detection of dopamine and glucose in sweat. In this protocol, one of the working electrodes was used for dopamine (DA) sensing, and the other working electrode was used for glucose sensing. The prepared four-electrode electrochemical sensor was screen-printed on flexible polydimethylsiloxane (PDMS) substrate, which allows the sensor to fit comfortably on curved skin surfaces. The prepared wearable patch sensor was finally attached to skin for simultaneous electrochemical detection of dopamine and glucose in sweat.

## 2. Experimental Section

### 2.1. Chemicals and Materials

$Na_2HPO_4 \cdot 12H_2O$ ($\geq$99%), $NaH_2PO_4 \cdot 2H_2O$ ($\geq$99%), KCl ($\geq$99.5%), $K_3Fe(CN)_6$ ($\geq$99.5%), $K_4Fe(CN)_6 \cdot 3H_2O$ ($\geq$99.5%), $H_2O_2$ (30%), acetic acid ($\geq$99.8%), ascorbic acid ($\geq$99.7%), lactic acid (85–92%), urea ($\geq$99%) and D(+)-glucose anhydrous were purchased from Sinopharm Chemical Reagent Co., Ltd. (Shanghai, China). Nafion (5 wt%) and chitosan (medium molecular weight) were obtained from Sigma-Aldrich (Shanghai, China). Carbon mediator paste (Prussian blue) and silver/silver chloride (60:40) inks were purchased from Sun-Chemical Co. (Parsippany, NJ, USA). The carbon ink was purchased from Jujo Printing Supplies&Technology Co., Ltd. (Pinghu, China). 3-hydroxytyramine hydrochloride ($\geq$99%) was purchased from Shanghai Titan Scientific Co., Ltd. (Shanghai, China). Glucose oxidase (GOD, $\geq$100 U/mg) was obtained from Sangon Biotech (Shanghai, China). Uric acid ($\geq$98%) was obtained from TCI Shanghai. Carboxyl functionalized multi-walled carbon nanotubes (MWCNT-COOH) ($\geq$98 wt%) was purchased from Chengdu Organic Chemicals Co., Ltd. (Chengdu, China). SYLGARD 184 silicone elastomer kit was obtained from Dow silicones Co., Ltd. (Midland, MI, USA). A 0.1 M phosphate-buffered solution (PBS), pH 7.4, was prepared using $Na_2HPO_4 \cdot 12H_2O$, $NaH_2PO_4 \cdot 2H_2O$ and 0.1 M KCl. All chemicals were used without further purification. Deionized water from a Milli-Q system (18.2 M$\Omega$ cm at 25 °C) was used throughout the experiments.

### 2.2. Apparatus

All the electrochemical experiments were carried out using a CHI-760e workstation (CH Instruments, Shanghai, China). The electrode printing stencil was cut out on a vinyl transfer film using the Roland desktop cutter GS-24. Then, the electrodes were printed on the flexible PDMS substrate.

### 2.3. Preparation of PDMS Substrate

For preparation of the flexible PDMS substrate, a cuboid groove with a length of 20 mm, width of 15 mm and height of 0.3 mm was first prepared. The groove was made of polyethylene terephthalate (PET) as the base and tape as the walls. Then, a mixture of part A (SYLGARD 184 silicone elastomer kit) mixing with part B (SYLGARD 184 silicone elastomer kit) at a mass ratio of 10:1 was poured evenly into the groove. Subsequently, the entire setup was put in vacuum oven at 75 °C for 40 min. Finally, a flexible PDMS substrate was obtained and the tape was removed before use. The schematic diagram of PDMS substrate preparation process can be seen in Figure 1.

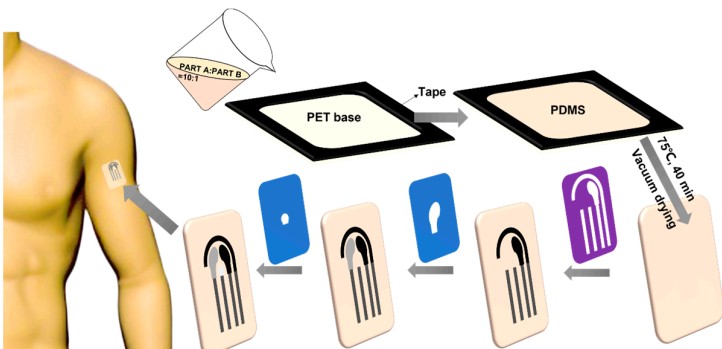

**Figure 1.** The schematic diagram of PDMS substrate preparation and electrode printing process.

### 2.4. Printing of the Electrodes

The electrode printing process is similar to that in our previous report but with a few modifications [3]. In detail, the electrode printing stencil was designed using Adobe Illustrator software before printing. Then, the vinyl stencil was cut out using the Roland desktop cutter. Subsequently, the vinyl stencil transfer films were progressively pasted and removed after the corresponding inks were progressively applied on the flexible PDMS substrate. It should be pointed out that, after each kind of ink was applied, the electrode setup was put in oven at 60 °C for 10 min. The printed four electrodes consisted of two working electrodes of Prussian blue (PB) modified carbon electrode (WE-PB) and unmodified carbon electrode (bare WE), reference electrode (RE) of Ag/AgCl, and counter electrode (CE) of unmodified carbon, respectively. The schematic diagram of the electrodes printing process can be found in Figure 1.

### 2.5. Preparation of Dopamine Biosensor (Sensor-DA)

The sensor for DA sensing was prepared by modifying MWCNT-COOH on screen-printing carbon electrode (SPCE). In detail, 1 mg MWCNT-COOH was mixed with 1 mL 2 wt% Nafion solution, which was previously prepared by diluting 5 wt% Nafion stock in deionized water. Then, the mixture was treated by ultrasound for half an hour. Finally, 3 µL MWCNT-COOH/Nafion composite was drop-cast on bare WE.

### 2.6. Preparation of Glucose Biosensor (Sensor-Glucose)

Firstly, the MWCNT-COOH/chitosan solution was prepared by dispersing 2 mg MWCNT-COOH in 1 mL 1 wt% acetic acid and 2 mL chitosan solution. Then, the MWCNT-COOH/chitosan solution was mixed with 40 mg/mL GOD solution, which was prepared by dissolving GOD in PBS, at a volume ratio of 2:1. Finally, 3 µL GOD/MWCNT-COOH/chitosan composite was drop-cast on WE-PB and stored in a fridge at 4 °C overnight.

### 2.7. Patch Sensor Design and Electrochemical Sensing Mechanism

The prepared wearable patch sensor can be tightly attached to the skin for continuous monitoring of biomarkers in sweat, as shown in Figure 2A. Figure 2B indicates the sensor's design sizes. This four-electrode patch sensing system was printed on a flexible PDMS membrane with a simple screen-printing technique. Figure 2C shows the digital photo of the patch sensor, and it resembles a "panda" shape, as shown in Figure 2D.

The electrochemical sensing mechanism of the wearable patch sensor is based on the enzymatic reaction of glucose oxidized on GOD-modified electrode in the presence of oxygen integrating with the electrochemical reaction of hydrogen peroxide reduced on a PB-modified carbon electrode. On one of these working electrodes (Sensor-glucose), hydrogen peroxide was produced enzymatically by GOD, and its concentration was proportional to glucose concentration. Then, hydrogen peroxide was electrochemically reduced on WE-PB. On the other working electrode (Sensor-DA), DA was electrochemically oxidized

on the MWCNT-COOH-modified carbon electrode to produce dopaminequinone by a two-electron reaction process [31]. The modification process and electrochemical sensing mechanism of the wearable patch sensor are illustrated in Figure 3.

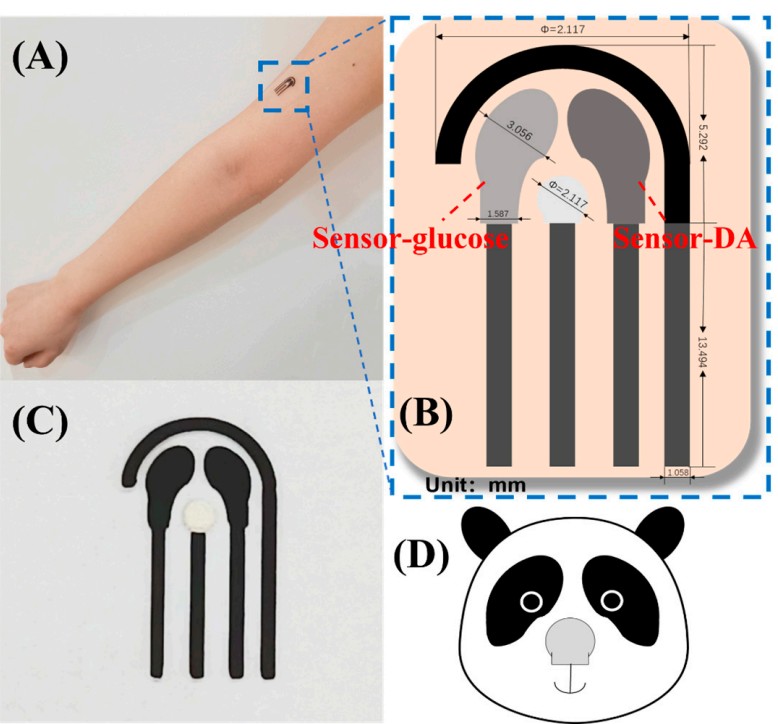

**Figure 2.** (**A**) The prepared patch sensor can be tightly attached to the skin for continuous monitoring of biomarkers in sweat. (**B**) Schematic design of the patch sensor and the design sizes are indicated. (**C**) The digital photo of the four-electrode patch sensor, and it resembles a (**D**) "panda" shape.

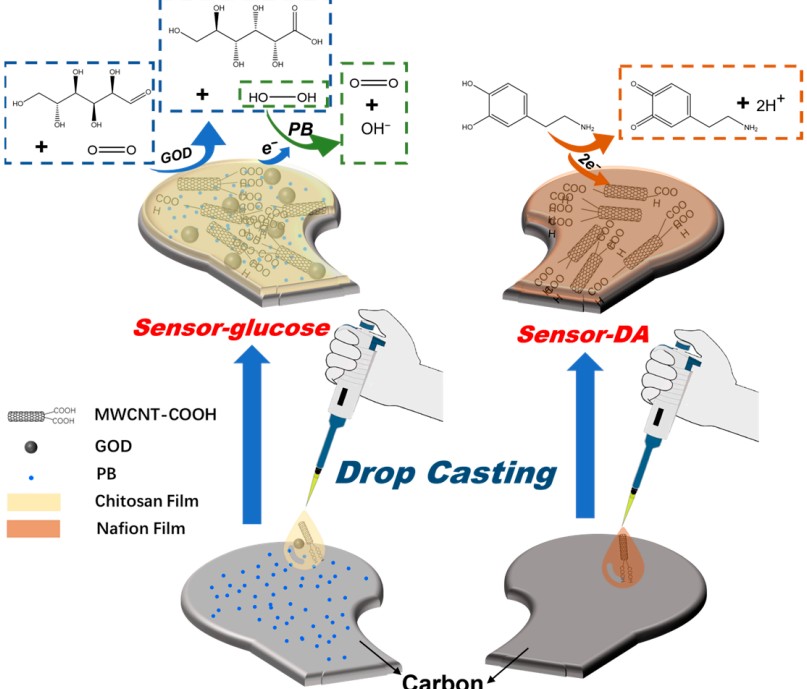

**Figure 3.** The modification process and electrochemical sensing mechanism of the wearable patch sensor for glucose and DA sensing.

## 3. Results and Discussion

### 3.1. Electrochemical Sensing Performance of Sensor-DA toward Dopamine

To investigate the electron-transfer property of SPCE after the modification of MWC-NTs, electrochemical impedance spectroscopy (EIS) was carried out in 0.1 M KCl solution containing 5 mM $KFe^{II}[Fe^{III}(CN)_6]$. The potential was controlled at the open circuit potential (OCP) of 0.17 V. As Figure 4A shows, two classic EIS Nyquist plots of SPCE and Sensor-DA were obtained, and the semicircle plots in high-frequency region corresponded to the dynamic process. It is obvious that, after the modification of MWCNTs on SPCE, the interfacial electron-transfer resistance ($R_{et}$) increases. This is due to the formation of different kinetic barriers after the modification, resulting in the increase in $R_{et}$ [5]. This indicates the successful modification of MWCNTs on SPCE. Moreover, the electrochemical sensing performance of the Sensor-DA toward DA was investigated using cyclic voltammetry (CV), which was performed in 0.1 M PBS, pH 7.4, at a scan rate of 50 mV/s, potential ranging from −0.35 V to 0.8 V (vs. Ag/AgCl). The electrochemical cyclic voltammograms of bare WE and Sensor-DA are shown in Figure 4B. The inset showed the cyclic voltammograms of the bare WE in absence and presence of 80 μM DA. In the absence of DA, the CV of bare WE showed classic cyclic voltammogram of carbon in PBS with no obvious redox peak. After the addition of 80 μM DA, an anodic peak at 0.5 V and a cathodic peak at −0.1 V were found. These redox peaks should be ascribed to the electrochemical redox reactions between DA and dopaminequinone of a two-electron reaction process [31,32]. Moreover, compared to bare WE, after the modification of MWCNT-COOH, Sensor-DA showed an increased double layer capacitance in PBS without the presence of DA. It revealed the successful modification of MWCNT-COOH on bare WE. In the presence of 80 μM DA, Sensor-DA showed a pair of well-defined redox peaks, where the anodic peak was at 0.14 V and the cathodic one was at 0.01 V. The above result showed that, compared to bare WE, Sensor-DA characterized lower redox overpotential and more comparable redox peaks current value toward the electrochemical reaction of DA in PBS. It showed the enhanced electrocatalytic activity of MWCNT-COOH compared to unmodified carbon electrode. Figure 4C showed the CVs of Sensor-DA toward 0, 20, 40, 60, and 80 μM DA in 0.1 M PBS, pH 7.4. The well-defined redox peaks current increased with the increase in DA concentrations, showing Sensor-DA's potential electrochemical sensing ability. To further investigate the sensing performance of Sensor-DA toward DA, differential pulse voltammetry (DPV) was applied in 0.1 M PBS, pH 7.4, containing different concentrations of DA. The parameters of DPV were the potential range of −0.1 V to 0.4 V, the amplitude of 0.05 V, the pulse width of 0.05 s, and the pulse period of 0.5 s. As Figure 4D illustrated, the DPV anodic peak current increased with the increase in DA concentrations. The inset showed the calibration curve of the relationship between response peak current against DA concentration. The limit of detection (LOD) of Sensor-DA toward DA was calculated as 0.043 μM (S/N = 3). Moreover, the curve could be divided into two parts. At low concentration of DA, the electrocatalytic oxidation mechanism is dominant; yet at high concentration of DA, the ability of surface electro-oxidation is crucial for current response [33]. The linear ranges for these two calibration curves were from 0 to 70 μM with the linear correlation coefficients ($R^2$) of 0.9938, and 70 μM to 180 μM with the $R^2$ of 0.9961, respectively. The equations for these two linear fitting curves are as follows:

$$I\ (\mu A) = 0.6740 \times c(DA)(\mu M) + 26.0762\ (R^2 = 0.9938),$$

and

$$I\ (\mu A) = 0.2290 \times c(DA)(\mu M) + 56.9986\ (R^2 = 0.9961),$$

respectively.

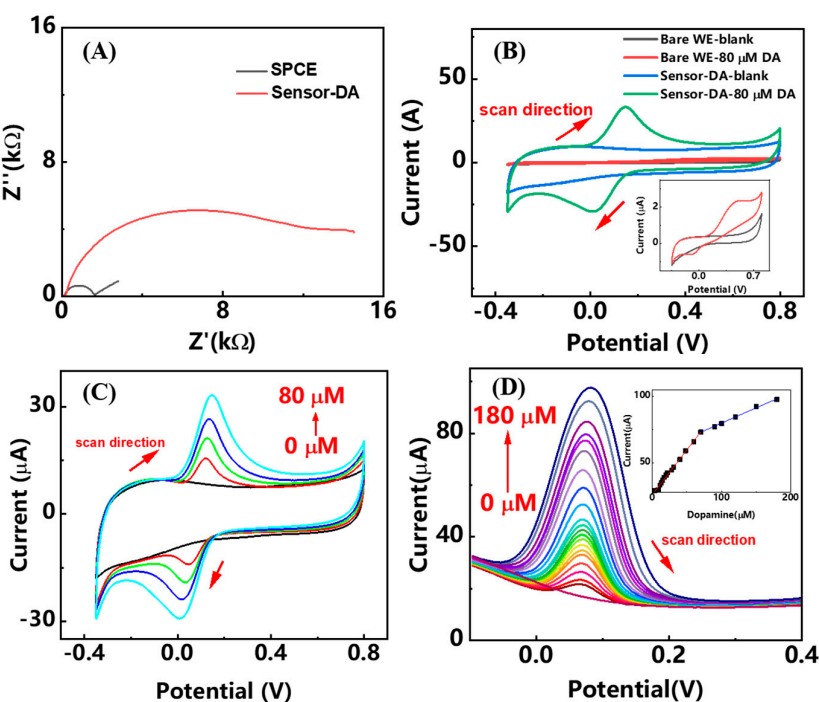

**Figure 4.** The electrochemical sensing performance of Sensor-DA toward dopamine. (**A**) EIS of SPCE and Sensor-DA in 0.1 M KCl containing 5 mM KFe$^{II}$[Fe$^{III}$(CN)$_6$]. The potential was controlled at the open circuit potential (OCP) of 0.17 V. (**B**) Cyclic voltammograms of bare WE and Sensor-DA in PBS, pH 7.4, with or without 80 μM DA. Scan rate 50 mV/s, potential range −0.35 V to 0.8 V (vs. Ag/AgCl). The inset shows the zoomed in cyclic voltammograms of bare WE in PBS with or without 80 μM DA. (**C**) Cyclic voltammograms of Sensor-DA toward the different concentrations of DA. Scan rate 50 mV/s, potential range −0.35 V to 0.8 V (vs. Ag/AgCl). (**D**) Differential pulse voltammograms of Sensor-DA in PBS containing different concentrations of DA. Potential range −0.1 V to 0.4 V, amplitude 0.05 V, pulse width 0.05 s, pulse period 0.5 s. The inset shows the calibration curve of the relationship between response peaks current against DA concentration.

### 3.2. Electrochemical Sensing Performance for Glucose

The electrochemical sensing performance of Sensor-glucose was investigated in 0.1 M PBS, pH 7.4. The electrochemical sensing mechanism of Sensor-glucose is based on the enzymatic reaction of glucose by GOD. Glucose can be enzymatically oxidized in the presence of oxygen and GOD to produce hydrogen peroxide. Then, hydrogen peroxide can be further electrochemically reduced on the PB-modified electrode [34]. The current value of the final response is proportional to the glucose concentration. Thus, the electrochemical sensing performance of the PB-modified carbon electrode was investigated first. As Figure 5A shows, WE-PB showed a pair of well-defined redox potential of PB in PBS solution. The anodic peak was at 0.13 V and the cathodic one was at 0.06 V. It is needed to be pointed out that the redox peak difference (around 0.7 V) is close to the theoretical reversible value of one electron transport. This should be ascribed to the reversible electrochemical redox reaction for interconversion between Prussian blue and Prussian white [35]. As hydrogen peroxide concentration increased from 0.4 to 1 mM, the electrochemical reduction peak current increased. This showed the electrochemical sensing ability of PB-modified carbon electrode toward hydrogen peroxide. Figure 5B shows the chronoamperometry response of WE-PB to the different concentration of hydrogen peroxide from 0 to 5.1 mM in 0.1 M PBS, pH 7.4. A constant potential of 0.1 V was applied for 60 s, where the current response value was read. As hydrogen peroxide concentrations increased, the electrochemical reduction current increased. Correspondingly, the calibration curve was obtained as shown in the

inset. The linear range of WE-PB sensor toward hydrogen peroxide was from 0 to 3.6 mM with the LOD of 11.46 µM (S/N = 3). The linear fitting equation is as follows:

$$I\ (\mu A) = -2.5065 \times c(H_2O_2)\ (mM) - 0.2614\ (R^2 = 0.9934).$$

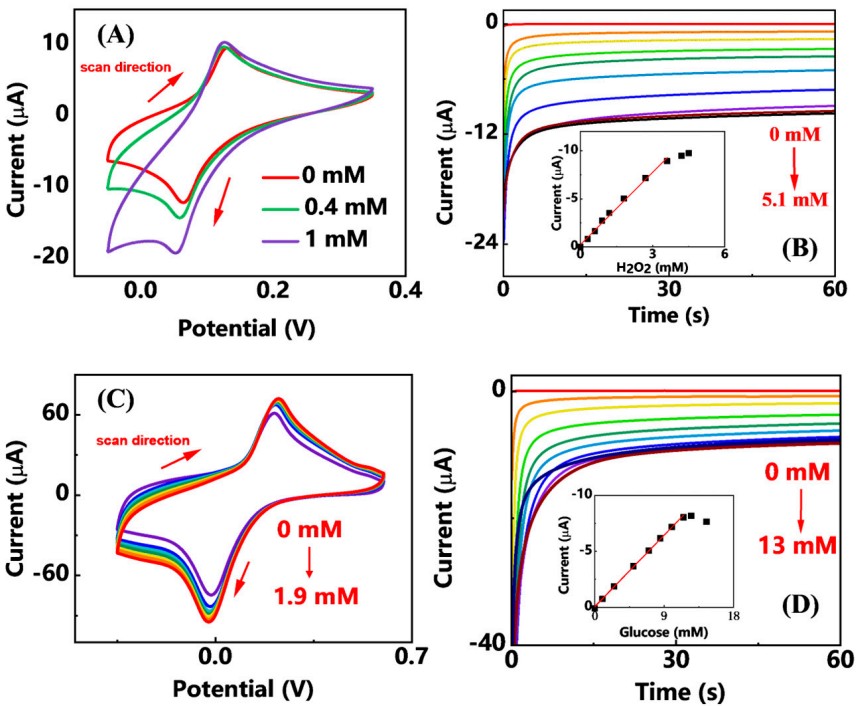

**Figure 5.** The electrochemical sensing performance of Sensor-glucose toward glucose. (**A**) Cyclic voltammograms of WE-PB in the different concentrations of $H_2O_2$, in 0.1 M PBS, pH 7.4. Scan rate 50 mV/s, potential range −0.05 to 0.35 V (vs. Ag/AgCl). (**B**) Chronoamperometric response of WE-PB to the different concentrations of $H_2O_2$, in 0.1 M PBS, pH 7.4. Potential 0.1 V (vs. Ag/AgCl), time 60 s. The inset shows its corresponding calibration curve. (**C**) Cyclic voltammograms of Sensor-glucose in the different concentrations of glucose, in 0.1 M PBS, pH 7.4. Scan rate 50 mV/s, potential range −0.05 to 0.35 V (vs. Ag/AgCl). (**D**) Chronoamperometric response of Sensor-glucose to the different concentrations of glucose, in 0.1 M PBS, pH 7.4. Potential 0.1 V (vs. Ag/AgCl), time 60 s. The inset shows its corresponding calibration curve.

To investigate the electrochemical sensing performance of Sensor-glucose, CVs of Sensor-glucose in 0.1 M PBS, pH 7.4 were performed. As Figure 5C shows, the electrochemical reduction current increased when the glucose concentration increased from 0 to 1.9 mM. This showed the electrochemical sensing ability of Sensor-glucose toward glucose. To further investigate its sensing performance, chronoamperometry was carried out in 0.1 M PBS, pH 7.4, as shown in Figure 5D. A constant potential of 0.1 V was applied for 60 s, where the current response value was read. As glucose concentrations increased from 0 to 13 mM, the electrochemical reduction current increased. The corresponding calibration curve was obtained as shown in the inset. The linear range of Sensor-glucose toward glucose was from 0 to 11.5 mM with the LOD of 40.36 µM (S/N = 3). The linear fitting equation is as follows:

$$I\ (\mu A) = -0.7117 \times c(glucose)\ (mM) - 0.0322 (R^2 = 0.9989).$$

### 3.3. Anti-Interference Ability, Reproducibility and Stability

Sweat contains a rich variety of biomarkers, such as glucose, dopamine, uric acid, urea, lactic acid, ascorbic acid, and so on. Many of these compounds are also electrochemically active. Anti-interference ability is also a key indicator for evaluating sensor performance. To

evaluate the anti-interference ability of the patch sensor, chronoamperometry was carried out. Figure 6A shows the Sensor-DA's chronoamperometric response toward combinations of dopamine and different species. A constant potential of 0 V (vs.Ag/AgCl) was controlled for 60 s, and the current response against time was recorded. It can be found that the Sensor-DA provided a current response around 0.3 μA to 20 μM DA. However, compared to a single 20 μM DA, the Sensor-DA provided a similar current response to the compositions containing 20 μM DA and different interferences. This indicated that the Sensor-DA has a good anti-interference ability. Similarly, the interference ability of the Sensor-glucose was investigated by chronoamperometric test. A constant potential of 0.1 V (vs.Ag/AgCl) was controlled for 60 s. As Figure 6B shows, the Sensor-glucose provided a 1.23 μA current response to 300 μM glucose, but a similar response to compositions containing 300 μM glucose and different interferences. This also indicated that the Sensor-glucose has a good anti-interference ability.

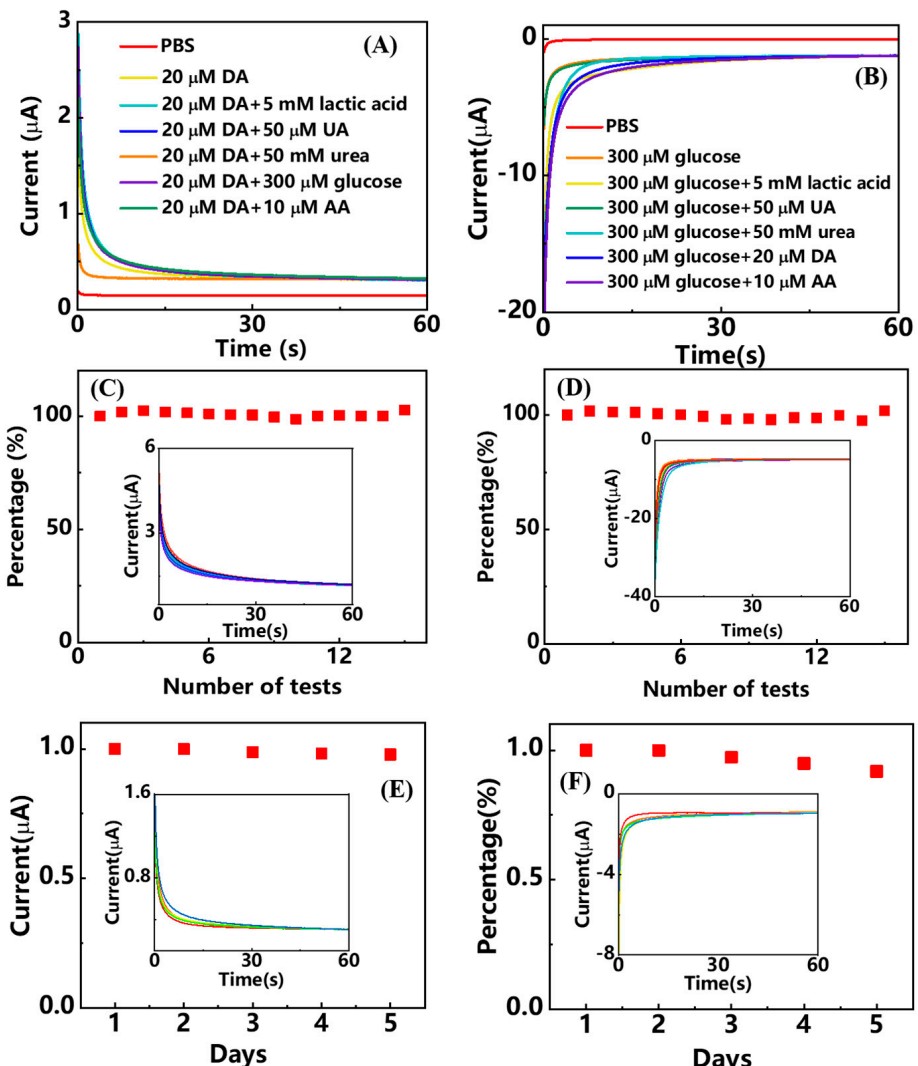

**Figure 6.** The anti-interference ability and reproducibility tests in 0.1 M PBS, pH 7.4. (**A**,**B**) are anti-interference tests of Sensor-DA and Sensor-glucose. The potentials were applied at 0 and 0.1 V (vs.Ag/AgCl), respectively, for 60 s. Chronoamperometric response toward combinations of analyte (DA or glucose) and different species were recorded. (**C**,**D**) are the reproducibility tests of Sensor-DA and Sensor-glucose. The current response percentage of the 15 tests relative to the first one was recorded. The insets show the respective chronoamperometric response. (**E**,**F**) are the stability tests of Sensor-DA and Sensor-glucose. The insets show the corresponding chronoamperometric responses in PBS solution containing 20 μM DA and 500 μM glucose, respectively.

To investigate the reproducibility of the patch sensor, 15 chronoamperometric tests were performed on Sensor-DA and Sensor-glucose in PBS solutions containing 70 μM DA and 5 mM glucose, respectively. The applied constant potentials were controlled at 0 V and 0.1 V, respectively. Figure 6C shows the current response percentage of Sensor-DA of the 15 tests relative to the first one, and the inset shows the recorded 15 chronoamperometric tests. The relative standard deviation (RSD) of the 15 test response values was 1.184%. Similarly, Figure 6D shows the current response percentage of Sensor-glucose of the 15 tests relative to the first one, and the inset shows the recorded 15 chronoamperometric tests. The response revealed an RSD of 1.443%. These results showed that there was no significant difference in the 15 chronoamperometric response values of the patch sensor, demonstrating good reproducibility.

The stability of the wearable electrochemical patch sensor was also evaluated for five days. As shown in Figure 6E,F, stability tests of the patch sensor were performed daily for five days, in a PBS solution containing, respectively, 20 μM DA and 500 μM glucose. The insets show the corresponding chronoamperometric responses. It can be found that, compared to the first day, the current response of the Sensor-DA has almost no attenuation. This is ascribed to the stability of the inorganic material of carbon ink and carbon nanotubes. However, the current response of the Sensor-glucose decreased at day 5, but remained at 91.9%. This is due to the intrinsic instability of biomaterials of glucose oxidase.

### 3.4. Electrochemical Sensing in Sweat

The evaluation of real sample sensing performance of the patch sensor was carried out in sweat, which was collected by a volunteer after exercise. DPV and chronoamperometry were applied for Sensor-DA and Sensor-glucose, respectively. The electrochemical parameters of the evaluations were the same with the ones for the patch sensor in PBS. Figure 7A shows that the DPV peak current of Sensor-DA increased with the increase in DA concentrations, and the inset shows the corresponding calibration curve. Similar to the case in PBS, it can be found that the calibration curve was split into two parts within the linear ranges from 0 to 50 μM and 50 μM to 120 μM, respectively. An LOD of 0.065 μM was obtained (S/N), and these two linear equations followed:

$$I\ (\mu A) = 0.4397 \times c(DA)(\mu M) + 7.6942\ (R^2 = 0.9924),$$

and

$$I\ (\mu A) = 0.1865 \times c(DA)(\mu M) + 19.3035\ (R^2 = 0.9969),$$

respectively.

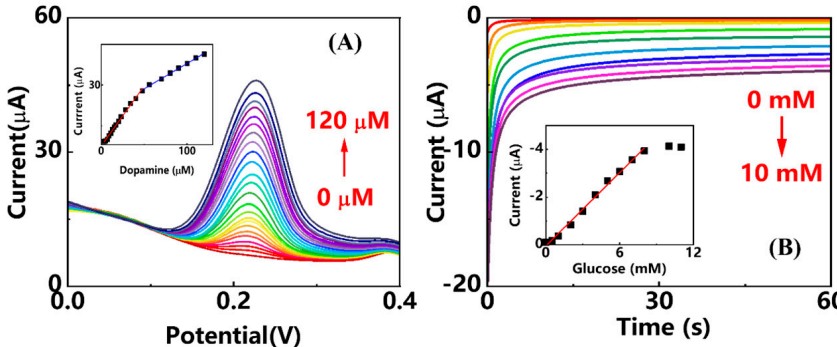

**Figure 7.** The electrochemical sensing performance of the wearable patch sensing in sweat. (**A**) Differential pulse voltammograms of Sensor-DA toward different concentrations of DA. The inset shows the corresponding calibration curve. Potential range 0 to 0.4 V, amplitude 0.05 V, pulse width 0.05 s, pulse period 0.5 s. (**B**) Chronoamperometric response of Sensor-glucose to different concentrations of glucose. Potential 0.1 V(vs.Ag/AgCl), time 60 s. The inset shows the corresponding calibration curve.

Figure 7B shows the chronoamperometric current response of Sensor-glucose to different concentrations of glucose in sweat. In addition, the electrochemical reduction current increased with the glucose concentration increasing. The inset shows the corresponding calibration curve, and the linear range was from 0 to 8 mM. The LOD was calculated as 55.65 µM. The linear fitting equation is as follows:

$$I\ (\mu A) = -0.5124 \times c(glucose)(mM) + 0.0428 (R^2 = 0.9941).$$

It is worth mentioning that the linear detection range of this wearable electrochemical patch sensor for glucose covers well its physiological concentration range in sweat from 0.01 to 1.11 mM [36]. The comparison of the electrochemical sensing performance of the wearable patch sensor with other reported ones for dopamine and glucose sensing in sweat is listed in Table 1. It demonstrates good electrochemical sensing performance of the wearable patch sensor.

**Table 1.** Comparisons of the electrochemical sensing performance of the modified electrodes for dopamine or glucose sensing in sweat.

| Electrode | Modifications | Biomarker | LOD (µM) | Linear Range (µM) | Reference |
|---|---|---|---|---|---|
| Glassy carbon | CuO-MgO | Dopamine | 6.4 | 10–100 | [23] |
| Graphite paper | Mn-MoS$_2$ | Dopamine (in artificial sweat) | 0.05 | 0.05–500 | [24] |
| Carbon nanotube/cellulose nanocrystal | Dopamine-imprinted PANI-co-PBA | Dopamine | 0.00211 | 0–0.763 | [37] |
| SPCE | MWCNT-COOH | Dopamine | 0.065 | 0–50; 50–120 | This work |
| SPCE | GOD/PB-PEDOT | Glucose | 4 | 6.25–800 | [16] |
| 3D-PMED | GOD | Glucose | 5 | 0–1900 | [17] |
| Kel-F fluorocarbon | Gold rod | Glucose | 15 | 30–1000 | [38] |
| SPCE | GOD/MWCNT-COOH/PB | Glucose | 55.65 | 0–8000 | This work |

## 4. Conclusions

In this work, we fabricated a wearable electrochemical patch sensor for simultaneous detection of dopamine and glucose in sweat. The prepared sensor was printed on flexible PDMS substrate, which can fit comfortably on curved skin surfaces. The Sensor-DA was prepared by the modification of MWCNT-COOH on screen-printed carbon electrode, exhibiting an enhanced dopamine electrochemical sensing performance compared to the unmodified electrode. The Sensor-glucose was prepared by immobilizing glucose oxidase on Prussian blue modified screen-printed carbon electrode, exhibiting high sensing sensitivity and specificity toward glucose electrochemical sensing. Moreover, the wearable patch sensor's electrochemical performance was fully evaluated in PBS, and it exhibited good sensitivity, wide linear range, low limit of detection, good anti-interference ability and reproducibility. We believe that the wearable electrochemical patch sensor's sensing performance, such as the limit of detection, can be further improved after integrating other functional materials. Finally, the wearable patch sensor was evaluated for dopamine and glucose electrochemical sensing in real sweat, demonstrating its potential application in health management and diagnosis of these two-compound-related diseases.

**Author Contributions:** Conceptualization, Z.L.; methodology, Z.L., Z.W. and Y.H.; software, Y.S. and J.M.; validation, Y.S.; formal analysis, Y.S.; investigation, Y.S. and Y.W.; resources, S.Q. and Y.F.; data curation, Y.S.; writing—original draft preparation, Z.L.; writing—review and editing, Z.Z.; visualization, J.M.; supervision, Z.L.; project administration, Z.Z.; funding acquisition, Z.Z. All authors have read and agreed to the published version of the manuscript.

**Funding:** This research was funded by the University Capacity Building Project of Shanghai Science and Technology Commission grant number 21010502800.

**Institutional Review Board Statement:** Not applicable.

**Informed Consent Statement:** Not applicable.

**Data Availability Statement:** Not applicable.

**Conflicts of Interest:** The authors declare no conflict of interest.

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
