# Peer review of "A Wearable Patch Sensor for Simultaneous Detection of Dopamine and Glucose in Sweat"

_analytica, doi:10.3390/analytica4020014_

Round 1

Reviewer 1 Report

Journal: Analytica

Manuscript Number: analytica-2359393   

Full Title: A wearable patch sensor for simultaneous detection of dopamine and glucose in sweat.

Article Type: Article.

In this paper (analytica-2359393), the experiment exploration is insufficient, the data processing is not rigorous, the logic is chaotic and not reflects the significance of the composite material development.

I recommend accepting the paper after a revision.

1-      What is the innovation of this article? Please highlight it in the article.

2-      The stability and homogeneity should be discussed in more detail for a glass samples.

3-      ‎‎How could the choice of preparing this sensor-glucose be justified scientifically for the purpose of the electrochemical sensing performance? Please describe the selectivity of the Anti-interference ability and reproducibility tests for the proposed process?

4-      Data in Fig. 4 and Fig. 5 are important, but these data were just presented only with short discussions. Discussions of mechanisms for these data have to be strengthened.

5-      The background of materials properties in the manuscript need to be enhanced, related literature:

https://doi.org/10.1007/s10854-019-01102-9          

 https://doi.org/10.1007/s10854-021-06888-1

 https://doi.org/10.1007/s10854-021-06268-9

6-      This work should be compared with recently published articles (at least 4-6) in a scientific table.

7-      The reported of electrochemical sensing performance of Sensor-DA toward dopamine (Fig.4), is not in correlation with other references. They must add references.

8-      More and proper discussion about the results it is necessary.

9-      The Authors should also proofread their manuscript (some spelling and grammar errors). 

1-      The Authors should also proofread their manuscript (some spelling and grammar errors). 

Reviewer 2 Report

The paper "A wearable patch sensor for simultaneous detection of dopamine and glucose in sweat" thoroughly investigates the development of a promising platform for electrochemical electrodes. The publication can be recommended, but a major revision is suggested. Detailed comments are provided below for the author. 

Comments: 

1. The author needs to include the EIS analysis. 

2. Please insert the direction of the scan in the voltammogram (Fig. 4–6).

3. The author needs to include the FTIR and Raman spectra.

4. There is Figure 6 in the manuscript. The authors are advised to recheck the manuscript correctly.

5. The authors need to include more literature in a Table format, which you need to prove that your proposed platform is superior to previously reported literature.

6. Authors should be trimmed/condensed in the revised manuscript's ‘Abstract’ and ‘Conclusion’ sections. Please keep highlights of the whole manuscript in both sections.

7. This manuscript has some spelling, typos, style errors, and grammatical errors, which severely affect its readability. So, I suggest the authors carefully check the whole manuscript and correct them

   This manuscript has some spelling, typos, style errors, and grammatical errors, which severely affect its readability. So, I suggest the authors carefully check the whole manuscript and correct them

Reviewer 3 Report

Article – “A wearable patch sensor for simultaneous detection of dopamine and glucose in sweat”

Authors – Yue Sun, Junjie Ma, Yuwei Wang, Sen Qiao, Zhanhong Li, Zifeng Wang, Yutong Han, Zhigang Zhu

Summary – The authors fabricated a PDMS based wearable patch with a four-electrode system to detect glucose and dopamine simultaneously. The authors claim that this patch can be used to detect both dopamine and glucose from sweat.

Overall, the work presented here has very low originality, the only addition is the design of a sensor with 4 electrodes that has 2 working electrodes functionalized to detect two different analytes. This article needs major revisions before it can be accepted for publication.

Comments-

1.     In Introduction, please cite references from other groups. There are plenty of recent advances in the field.

2.     Please check the relevance of reference 14, cited as a wearable device study.

3.     The important advancement here seems to be the simultaneous detection of two types of biomarkers. Could the authors please mention if there are any other advantages? For a wearable patch sensor, there are already much advanced designs available in the literature. For example, https://www.nature.com/articles/s41551-022-00916-z. Could the authors comment of using similar technology for glucose and dopamine sensing?

4.     In Figure 1, please label the electrodes for clarity.

5.     In Results and Discussion sections, the authors work with mM and 100s of µM concentrations of the analytes, however, the LODs reported are in 10s of nM range. Could the authors please include the process of determining LOD values? It is very difficult to see through the presented data.

6.     Could the authors please comment on the reusability, reproducibility in terms of different days of the patch?

Could the authors mention what are the range of concentrations aimed to be detected from sweat for both dopamine and glucose?

Major Comments-

7.     It will be important to detect naturally present dopamine and glucose concentrations from sweat samples to really demonstrate the applicability of this patch sensor. Could the authors perform such a study and include that in the article?

The authors claim that the sensor can detect both dopamine and glucose simultaneously. Could the authors demonstrate this claim by analyzing a solution containing both dopamine and glucose on a single patch sensor?

Round 2

Reviewer 1 Report

After authors' careful revision, the manuscript seems fluent and readable now. Most of the grammar mistakes were amended. The contents, discussions and the conclusions have been revised in this version. May be accepted for publication.

Author Response

We appreciate the reviewer's comment that the current version of the manuscript can be acceptable for publication in Analytica.

Reviewer 2 Report

Accepted in current format 

Author Response

(The authors gave the same response as above.)

Reviewer 3 Report

Either in discussion or conclusions section, could the authors add comments about any limitations of testing or future improvements in the design of the patch sensor. For example, sampling of real sweat samples and improving the LOD especially compared to the sensors available in literature.

Author Response

We thank the reviewer's comment. And we have added the discussion in Conclusion section. More details can be found in revised manuscript.

In this work, we have fabricated a wearable electrochemical patch sensor for simultaneous detection of dopamine and glucose in sweat. The prepared sensor was printed on flexible PDMS substrate, which can fit comfortably on curved skin surfaces. The Sensor-DA was prepared by the modification of MWCNT-COOH on screen-printed carbon electrode, exhibiting an enhanced dopamine electrochemical sensing performance compared to the unmodified electrode. The Sensor-glucose was prepared by immobilizing glucose oxidase on Prussian blue modified screen-printed carbon electrode, exhibiting high sensing sensitivity and specificity toward glucose electrochemical sensing. Moreover, the wearable patch sensor’s electrochemical performance was full evaluated in PBS, exhibiting good sensitivity, wide linear range, low limit of detection, good anti-interference ability and reproducibility. We believe that the wearable electrochemical patch sensor’s sensing performance, such as the limit of detection, can be further improved after integrating other functional materials. Finally, the wearable patch sensor was evaluated for dopamine and glucose electrochemical sensing in real sweat, demonstrating its potential application in health management and diagnosis of these two compounds related diseases.